# Polygraphic EEG Can Identify Asphyxiated Infants for Therapeutic Hypothermia and Predict Neurodevelopmental Outcomes

**DOI:** 10.3390/children9081194

**Published:** 2022-08-09

**Authors:** Licia Lugli, Isotta Guidotti, Marisa Pugliese, Maria Federica Roversi, Luca Bedetti, Elisa Della Casa Muttini, Francesca Cavalleri, Alessandra Todeschini, Maurilio Genovese, Luca Ori, Maria Amato, Francesca Miselli, Laura Lucaccioni, Natascia Bertoncelli, Francesco Candia, Tommaso Maura, Lorenzo Iughetti, Fabrizio Ferrari, Alberto Berardi

**Affiliations:** 1Neonatology Unit, Mother-Child Department, University Hospital of Modena, Via del Pozzo 71, 41100 Modena, Italy; 2Psychology Unit, University Hospital of Modena, 41100 Modena, Italy; 3PhD Program in Clinical and Experimental Medicine, University of Modena and Reggio Emilia, 41100 Modena, Italy; 4Neuroradiology Unit, University Hospital of Modena, 41100 Modena, Italy; 5Pediatric Unit, Mother-Child Department, University Hospital of Modena, 41100 Modena, Italy; 6Postgraduate School of Pediatrics, Department of Medical and Surgical Sciences for Mother, Children and Adults, University of Modena and Reggio Emilia, 41100 Modena, Italy

**Keywords:** hypoxic–ischemic encephalopathy, therapeutic hypothermia, EEG, neurodevelopmental outcome

## Abstract

**Background:** Neonatal encephalopathy due to perinatal asphyxia is one of the leading causes of neonatal death and morbidity worldwide. The neurodevelopmental outcomes of asphyxiated neonates have considerably improved after therapeutic hypothermia (TH). The current challenge is to identify all newborns with encephalopathy at risk of cerebral lesions and subsequent disability within 6 h of life and who may be within the window period for treatment with TH. This study evaluated the neurodevelopmental outcomes in surviving asphyxiated neonates who did and did not receive TH, based on clinical and polygraphic electroencephalographic (p-EEG) criteria. **Methods:** The study included 139 asphyxiated newborns divided into two groups: 82 who received TH and 57 who were not cooled. TH was administered to asphyxiated newborns (gestational age ≥ 35 weeks, birth weight ≥ 1800 g) with encephalopathy of any grade and moderate-to-severe p-EEG abnormalities or seizures. Neurodevelopmental outcomes between the groups at 24 months of life and the risk factors for severe outcomes were assessed. **Results:** Severe neurodevelopmental impairment occurred in 10 (7.2%) out of the 139 enrolled neonates. Nine out of the 82 cooled neonates (11.0%) had severe neurodevelopmental impairment. All but one neonate (98.2%) who did not receive TH had normal outcomes. The multivariate logistic regression analysis showed that abnormal p-EEG patterns (OR: 27.6; IC: 2.8–267.6) and general movements (OR: 3.2; IC: 1.0–10.0) were significantly associated with severe neurodevelopmental impairment (area under ROC curve: 92.7%). **Conclusion:** The combination of clinical and p-EEG evaluations in hypoxic–ischemic encephalopathy contributed to a more accurate selection of patients treated with therapeutic hypothermia. When administered to infants with moderate to severe p-EEG abnormalities, TH prevents approximately 90% of severe neurodevelopmental impairment after any grade of hypoxic–ischemic encephalopathy.

## 1. Introduction

Neonatal encephalopathy after perinatal asphyxia is one of the leading causes of neonatal death and morbidity worldwide; however, therapeutic hypothermia (TH) has significantly improved clinical outcomes [1]. Randomized clinical trials (RCT) have shown that TH is effective in reducing death and disability in term newborns with moderate to severe hypoxic-ischemic encephalopathy (HIE) [2,3,4,5]. Due to a lack of RCTs supporting the administration of TH to infants with mild HIE in the first 6 h of life, the full potential benefit of TH remains unclear. However, increasing evidence from cohort studies shows that some untreated neonates may develop disabilities [5,6,7]. Hence, the current challenge is to identify all newborns with encephalopathy at risk of cerebral lesions and subsequent disability within 6 h of life and who may be within the window period for treatment with TH. The extent of cerebral injury (basal ganglia–thalami or cortical watershed lesions) in HIE is not only determined by the biochemical cascades that trigger the apoptosis–necrosis continuum of cell death in the brain parenchyma, but also by the breaching of the blood–brain barrier by pro-inflammatory factors. Several studies showed the usefulness of neuro-biomarkers (IL-1, IL-10, TNF-alpha, glial fibrillary acidic protein, ubiquitin carboxyl-terminal hydrolase L1, S100B, neuron-specific enolase) in detecting brain injury and in monitoring asphyxiated infants treated by TH [8,9,10]. Given the evolving nature of neonatal encephalopathy, the severity of HIE cannot be assessed accurately through clinical evaluation alone. In contrast, a neurophysiologic assessment may contribute to better defining of the severity of cerebral dysfunction in the first hours of life. Many centers use amplitude-integrated EEG (aEEG), an easy-to-interpret, real-time tool that shows changes in brain activity over time. Early aEEG findings in moderate to severe HIE correlate well with short- and long-term outcomes [11,12]. Indeed, aEEG has been used in conjunction with neurological scoring to determine whether an infant has moderate/severe HIE and is, therefore, eligible for TH [1,13]. However, aEEG monitoring within 6 h of birth seems insufficient for predicting the outcome of infants with mild HIE.

Continuous polygraphic EEG monitoring (p-EEG) is the gold standard method for assessing neonatal brain activity. Although it requires many electrodes to be placed on the neonate’s head and specialist interpretation, before the era of TH, several studies had shown that EEG recording was highly predictive of neonatal outcomes [14,15]. Therefore, p-EEG and clinical assessment might more accurately identify asphyxiated neonates who would benefit from TH.

This study assessed the neurodevelopmental outcomes in cooled (administered TH) and non-cooled (not administered TH) asphyxiated infants based on clinical and p-EEG criteria. Furthermore, we evaluated risk factors for severe neurodevelopmental outcomes.

## 2. Methods

### 2.1. Inclusion Criteria

This prospective study included 139 surviving asphyxiated newborns (gestational age ≥ 35 weeks and birth weight ≥ 1800 g) admitted to the neonatal intensive care unit (NICU) of the University Hospital of Modena between 1 January 2009, and 31 December 2019. TH was performed based on clinical and p-EEG criteria, such as (1) intrapartum asphyxia, confirmed by at least one of the following criteria: 10 min Apgar score ≤ 5, ventilation with an endotracheal tube (or mask) for at least 10 min after birth, severe acidosis (defined as cord pH or any arterial/venous pH ≤ 7.0 or base deficit ≥ 12 mmol/L within 60 min of birth); (2) neonatal encephalopathy assessed within 1 h of birth; (3) moderate to severe p-EEG abnormalities or seizures confirmed by p-EEG recording [1,2,14,15,16,17,18,19]. Among the 139 neonates, 82 newborns who met the requisite criteria were cooled (TH group), and 57 who did not meet the criteria were not cooled (no-TH group). Perinatal data, HIE severity, p-EEG, seizures, cerebral MRI, FM, GMDS-R scales, and outcomes were compared between the two groups.

Neurological examination was performed during the first hour of life. Encephalopathy was classified as mild (hyperactivity, normal or increased tone, normal spontaneous movements and posture, tremors, exaggerated Moro reflex, no autonomic dysfunction), moderate (lethargy, reduced motility, distal flexion/complete extension, hypotonia, weak/incomplete primitive reflexes, myosis, bradycardia, periodic breathing), and severe (stupor or coma, decerebrated posture, absent motility, flaccid tone, absent reflexes, mydriatic/deviated/non-reactive pupils, apnea) according to the modified Sarnat and Sarnat criteria [1,2,14,15,16,17,18,19]. TH was administered to newborns with moderate or severe HIE and moderate or severe p-EEG anomalies, regardless of the severity of HIE (including mild HIE). Patients were cooled to a rectal temperature of 33.5 °C for 72 h (CritiCool MTRE, Charter Kontron, Milton Keynes, UK) and then slowly rewarmed (+0.5 °C/h). Cooled infants received fentanyl analgesia as previously reported [20].

Exclusion criteria were congenital malformations, chromosomal abnormalities, metabolic disorders, sepsis or central nervous system infections, different causes of asphyxia (i.e., sudden unexpected postnatal collapse), incomplete TH (less than 72 h), or incomplete neurological follow-up.

This study was approved by the local ethics committee (Prot. AOU 0011282/20). Informed consent was obtained from the parents of each neonate included in this study.

### 2.2. p-EEG Recording

p-EEG monitoring was started within the first 6 h of life, as soon as possible after admission, using the 10–20 system of electrodes, electromyography (EMG), electrooculogram, and pneumogram (EB Neuro Galileo, Florence, Italy). EEG recordings were evaluated at the bedside by one of three experienced neonatologists (F.F, L.Lug., and I.G.) for every patient enrolled. Early p-EEG findings (<6 h) were classified according to the grading system described by Murray et al.: grade 0 (normal p-EEG), grade 1 (normal/mild abnormalities), grade 2 (moderate abnormalities), grade 3 (severe abnormalities), and grade 4 (inactive p-EEG) [1] (Figure 1). p-EEG monitoring was continued during hypothermic treatment (72 h) and rewarming. p-EEG signals at 24, 48, and 72 h were classified according to the system described by Murray et al. [14]. A seizure was defined as a sudden, repetitive, stereotyped discharge lasting ≥10 s on two or more EEG channels [21,22] (Figure 1). Antiepileptic drugs (AEDs) were administered based on online p-EEG evaluations. According to the local protocol, phenytoin was administered as the first-line and midazolam as the second-line AED [21,22].

### 2.3. Brain Magnetic Resonance Imaging

Brain magnetic resonance imaging (MRI) was performed within 30 postnatal days. The infants were scanned using a Philips Intera 1.5-T MRI scanner (Philips Medical Systems, Best, The Netherlands). Conventional and diffusion-weighted MRI sequences were obtained. Three experienced neuroradiologists (F.C., A.T., M.G.) scored the scans according to previously published criteria. Five patterns of injury were identified: moderate/severe damage in the basal ganglia and thalami associated with moderate/severe white matter (WM) changes and cortical injury (pattern 1); damage in the basal ganglia and thalami associated with mild WM changes with or without cortical injury (pattern 2); focal thalamic lesion with or without cortical injury (pattern 3); predominant WM damage (moderate/severe) with or without cortical injury with or without mild basal ganglia and thalami changes (pattern 4); mild WM abnormalities with or without mild cortical changes but with normal basal ganglia and thalami, or normal imaging (pattern 5) [23].

### 2.4. Neurological Follow-Up

Follow-up assessments were performed by experienced neonatologists trained in developmental neurology (F.F., L.Luc., L.B., M.F.R., E.D.), a developmental psychologist (M.P.), and a physical therapist (N.B.). The follow-up schedule included serial evaluations (at 3, 6, 12, and 24 months of age) with an assessment of general movements at 12–14 weeks, standard neurologic examination according to the protocols of Amiel–Tison and Touwen’s criteria, and Griffiths Mental Developmental Scales (GMDS-R) [24,25,26,27,28,29]. GMDS-R (0–2 years) provides a global development quotient (DQ) of infants’ abilities with a mean of 100.5, a standard deviation (SD) of 11.8, and five subscale quotients (locomotor, eye and hand coordination, personal and social, hearing and language, and cognitive performance) [25].

### 2.5. Assessment of General Movements

General movements (GMs) are gross movements involving the whole body, evident from the fetal period until approximately five months of post-term age [26,27,28]. For this study, fidgety movements (FM) were evaluated at 12–14 weeks and classified as follows: normal fidgety movements (small-amplitude movements of moderate speed and variable acceleration of the neck, trunk, and limbs in all directions, continual in the awake infant except during crying), absent fidgety movements (absence of fidgety movements but the presence of other movements), and abnormal fidgety movements (fidgety-like movements with moderately or greatly exaggerated amplitude, speed, and jerkiness) [28].

### 2.6. Neurodevelopmental Outcome

Neurodevelopmental outcomes were classified as normal (absent neurological signs and DQ > 85), moderately abnormal (clumsiness, poor balance, DQ 70–85, hearing impairment with no amplification), or severely abnormal (cerebral palsy, DQ < 70, epilepsy, a severe sensorineural deficit like bilateral deafness, requiring bilateral hearing aids or unilateral or bilateral cochlear implants, or bilateral blindness with visual acuity < 6/60 in the better eye) [21,22]. Cerebral palsy (CP) was defined as spastic (diplegia, hemiplegia, or quadriplegia), dystonic, or athetoid [29].

## 3. Statistical Analysis

MedCalc 8 software for Windows was used for statistical analyses. Descriptive statistics included mean, standard deviation (SD), median, and interquartile range (IQR) for continuous variables, and frequencies or proportions for categorical variables.

The groups were compared using χ2 analysis for categorical variables. Analysis of variance and Mann–Whitney U tests were used for continuous variables when normally or not normally distributed, respectively. Several variables were evaluated in the univariate analysis as possible risk factors for severe neurodevelopmental outcomes. Multivariate logistic regression analysis was performed using the best subset regression and backward variable selection strategy (entry criteria = 0.05 and stay criteria = 0.1). The multivariate analysis final model (area under ROC curve: 92.68%) included two variables (p-EEG at 48 h of age and FM). Statistical significance was set at *p* < 0.05.

## 4. Results

Among 139 included neonates, 82 were cooled (TH group) and 57 were not (no-TH group) (Figure 2). Perinatal data, HIE severity, p-EEG, seizures, cerebral MRI, FM, GMDS-R scales, and outcomes of the two groups are shown in Table 1. Seventeen (12.2%) out of 139 neonates had any grade of neurodevelopmental impairment (moderate, n = 7; severe, n = 10). Newborns with severe outcomes had CP (n = 7, 70%; comprising quadriplegia (n = 5), hemiplegia (n = 1), and dystonic CP (n = 1), hearing loss (n = 2, 20%), and cognitive delay (n = 1, 10%). All five infants with quadriplegia also had early epilepsy (Figure 3).

### 4.1. Cooled Infants (TH Group)

Among 82 cooled infants, 45 had moderate (54.9%), and 23 had severe (28.0%) HIE. In addition, 14 neonates with mild encephalopathy were cooled because of moderate-to-severe p-EEG abnormalities. The neurodevelopmental outcomes of these 82 cooled neonates were normal in 66 cases (80.5%), whereas 16 (19.5%) had neurodevelopmental impairment (moderate, n = 7, 8.5%; severe, n = 9, 11%) (Table 2). Among 23 cooled newborns with severe HIE, eight (34.8%) had severe and two (34.8%) had moderately abnormal outcomes (Table 2). Among 45 newborns with moderate HIE, one (2.2%) had severe outcomes, and four (8.9%) had moderate neurological outcomes. One of 14 patients (7.1%) with mild HIE (presenting with severe early p-EEG abnormalities) showed a moderate outcome (Table 2). Seizures occurred in 30/82 (36.6%) neonates during TH, and all infants with severe outcomes presented with seizures.

Early p-EEG was severely abnormal (grade 3) or inactive (grade 4) in all patients who developed severe outcomes; grade 2 or 3 p-EEG abnormalities were confirmed up to 48 and 72 h in all neonates with severe outcomes (Figure 4). Among neonates with normal outcomes, early p-EEG was moderately abnormal in 42/66 (63.3%) and severe in 24/66 (36.4%); however, at 72 h, p-EEG improved to mild abnormalities in 52/66 cases (78.8%) (Table 3) (Figure 5). Figure 6 shows the p-EEG variation in patients with severe neurodevelopmental outcomes (Figure 6A) and normal or moderate outcomes (Figure 6B).

Cerebral lesions were severe (pattern 1 or 2) in seven of nine patients with severe outcomes (77.8%), while patterns 3 and 4 were found in the remaining two (22.2%). FM was abnormal (n = 1; 11.1%) or absent (n = 7; 77.8%) in all but one of the infants with severe outcomes. In the univariate analysis, p-EEG, seizures, MRI, encephalopathy severity, and FM were associated with severe neurodevelopmental outcomes. In the multivariate analysis, p-EEG at 48 h of life (OR: 27.6; IC: 2.8–267.6) and FM (OR: 3.2; IC: 1.0–10.0) remained associated with severe outcomes (Table 4). Table 5 shows the prognostic accuracy of p- EEG, MRI, encephalopathy, and FM for severe outcomes in infants treated with TH. p-EEG at 48 h presented the best prognostic accuracy (area under ROC curve: 92.0%).

### 4.2. Non-Cooled Infants (No TH Group)

Among 57 infants who did not undergo TH, none had seizures or severe cerebral lesions (pattern 1 or 2), and all but one (98.2%) presented normal outcomes at 24 months. The only case (1.8%) with a severe outcome (cognitive delay) presented no apparent hypoxic–ischemic brain lesions on MRI (pattern 5) (Table 1). Univariate analysis showed no association with severe outcomes.

### 4.3. Mild HIE

Table 6 compares the 14 cooled and 57 non-cooled infants with mild HIE. The neonates were cooled based on p-EEG abnormalities (moderate, n = 10; severe, n = 4). Apgar scores at the 5th and 10th minute of cooled infants were significantly lower than those of non-cooled infants. FM, neurodevelopmental outcome, GMDS-R DQ, and its subscales did not differ between cooled and non-cooled mild HIE.

## 5. Discussion

Although RCTs on TH show a huge improvement of neurodevelopmental outcome after moderate to severe HIE, the outcome after mild encephalopathy remains unclear [1,2,3,7,19]. In contrast to previous trials that enrolled patients to receive TH based on clinical and sometimes aEEG criteria [1,16], in this study, we administered TH to asphyxiated patients based on both clinical and p-EEG criteria and for any grade of encephalopathy (including mild) with moderate to severe EEG abnormalities. Overall, severe neurodevelopmental outcomes at 24 months occurred in a small proportion of all infants with HIE (7.2%). Severe outcomes accounted for 11.0% of cooled infants and 13.2% of infants with moderate to severe HIE undergoing TH. This proportion aligns with our regional surveillance data [30] and is lower than that previously reported [1,2,3,31]. Although differences in study design and included population could play a role, our encouraging data may also be due to a more accurate selection of the asphyxiated infants to administer TH. Interestingly, patients with mild HIE who were not enrolled to receive TH (because of absent or mild EEG anomalies) presented normal outcomes in all but one case (98.2%).

As clinical evaluation alone may be elusive in assessing HIE severity soon after birth, neurophysiological recording can help identify apparently mild cases showing otherwise cerebral dysfunction on p-EEG (mild HIE plus). In infants with moderate-to-severe HIE, there is a good correlation between early aEEG findings and short- and long-term outcomes [11,12], but the ability to predict outcomes in infants with mild HIE appears to be limited. Most infants with mild HIE display either normal aEEG background activity or potentially slightly broader bands of activity, which can be subtle and difficult to detect [6]. In contrast, p-EEG can better discriminate the degree of abnormal cerebral activity even among patients with mild HIE. In a pre-hypothermia case-control study, Murray et al. showed that normal or mildly abnormal EEG < 24 h had a 100% positive predictive value for a normal outcome and a 70% negative predictive value at two years of age [14]. In one-third of cases, moderate EEG abnormalities appeared to be associated with moderate/severe cerebral injury, and intact survival at five years was reported in 46% of cases [14,31,32,33]. Therefore, we included patients with TH with moderate p-EEG abnormalities independent of HIE severity. Despite the lack of RCTs supporting the use of TH for mild HIE, many centers have reported a therapeutic creep, such that TH is now often offered to infants with mild HIE. For example, at a single site in Canada, among term newborns referred to the NICU for possible TH, 36% had mild HIE, and 16% of these infants received TH [34]. Analysis of the Children’s Hospital Neonatal Database, encompassing 27 regional NICUs in the United States, showed that of the 160 infants with mild HIE, 122 (76%) had received TH [35]. In a recent study of mild HIE, the rates of intact survival were comparable in controls, perinatal asphyxia without HIE, and mild HIE. There was no significant difference in Bayley Scales of Infant and Toddler Development composite scores between children with mild HIE treated with TH and non-treated children [7]. Consistent with these data, we found similar results when comparing cooled and non-cooled mild HIE.

Regarding prognosis in cooled patients, in our multivariate model, both p-EEG at 48 h of age and FM were associated with severe neurodevelopmental outcomes. Because p-EEG abnormalities tend to modify during TH, early EEG is not a good predictor of outcome, while p-EEG at 48 h better predicts prognosis. Severe EEG abnormalities persisted despite TH in infants who later showed severe disabilities. Regarding FM, several studies previously found that the quality of general movements is highly correlated with poor motor outcomes and central gray matter injury (patterns 1 and 2), the hallmark of acute perinatal asphyxia in full-term infants [28,36,37,38]. This study confirmed that FM is a good predictor of outcomes in cooled HIE infants. In contrast, in our study, cerebral MRI failed to correlate with neurodevelopmental outcomes in the multivariate logistic regression model, possibly because of the timing of the MRI. Cerebral MRI was performed over the first four postnatal weeks, a relatively long period. When MRI images were acquired very early, the conventional sequences could not be as reliable in predicting outcomes as when MRI was performed later. In fact, both overestimating brain injury due to transient abnormalities (such as cerebral edema) and underestimating the severity of damage can occur.

Our study has several limitations. First, reliable conclusions cannot be drawn because of the observational, non-randomized study design. Second, approximately 30% of infants were lost to follow-up, but this proportion is similar to that in previous studies [1,6,13]. Third, the follow-up duration was only 24 months, and some children classified with normal outcomes could later develop minor neurological or neuropsychological problems at the preschool or school age. In any case, cognitive and neurodevelopmental assessment at two years of life is a sensitive tool for the early identification of developmental impairment and delay, enabling the referral of high-risk children to early intervention. Finally, brain injury neuro-biomarkers were not evaluated, although their correlation with p-EEG remains to be established.

Nevertheless, this study is among the few to provide data on prolonged p-EEG during TH. Additionally, p-EEG has shown high diagnostic and prognostic accuracy by correctly selecting neonates to undergo TH and identifying those with poor outcomes. In fact, all infants with severe neurodevelopmental outcomes presented with early severe p-EEG abnormalities with no significant improvement during TH.

In conclusion, we selected to TH newborns with any grade of HIE and moderate to severe p-EEG abnormalities, thus including some mild HIE. Overall, severe neurodevelopmental disability occurred in less than 10% of cases, confirming the protective role of TH. Although untreated with TH, almost all neonates with mild p-EEG abnormalities, did not develop neurodevelopmental sequelae. The combination of clinical and p-EEG evaluations in HIE contributed to a more accurate selection of patients treated with TH. Furthermore, p-EEG at 48 h of age and general movements were the best predictors of severe neurodevelopmental impairment in cooled patients.

## Figures and Tables

**Figure 1 children-09-01194-f001:**
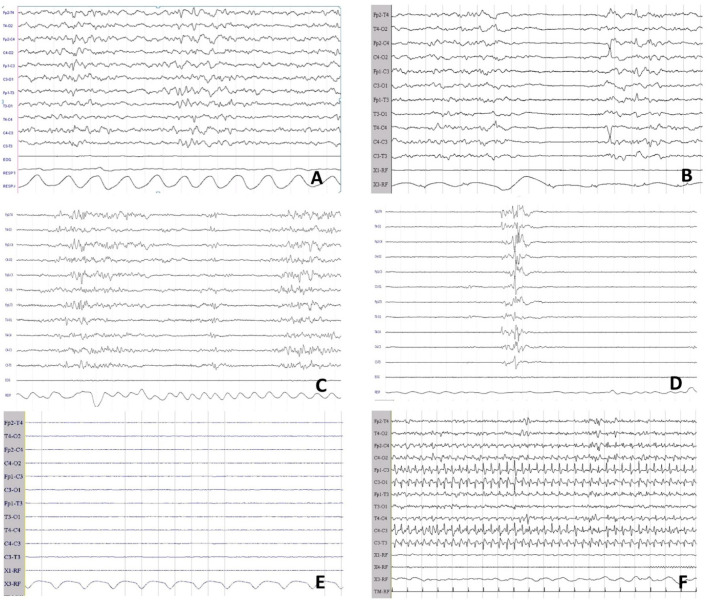
Classification of p-EEG abnormalities adapted with permission from Murray et al. [14]. p-EEG recording of patients included in the study are used as examples. (**A**): grade 0 (normal p-EEG). (**B**): grade 1 (normal/mild abnormalities). (**C**): grade 2 (moderate abnormalities). (**D**): grade 3 (severe abnormalities). (**E**): grade 4 (inactive p-EEG). (**F**): EEG confirmed seizure.

**Figure 2 children-09-01194-f002:**
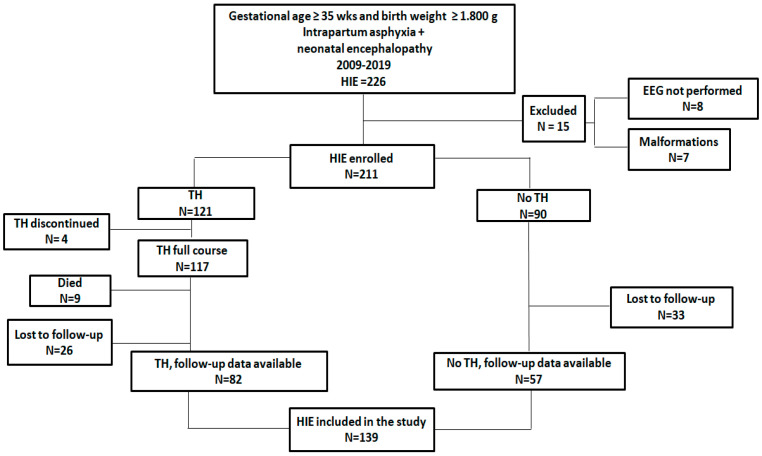
Flow_chart of asphyxiated infants undergoing or not undergoing therapeutic hypothermia. HIE: hypoxic–ischemic encephalopathy. TH: therapeutic hypothermia.

**Figure 3 children-09-01194-f003:**
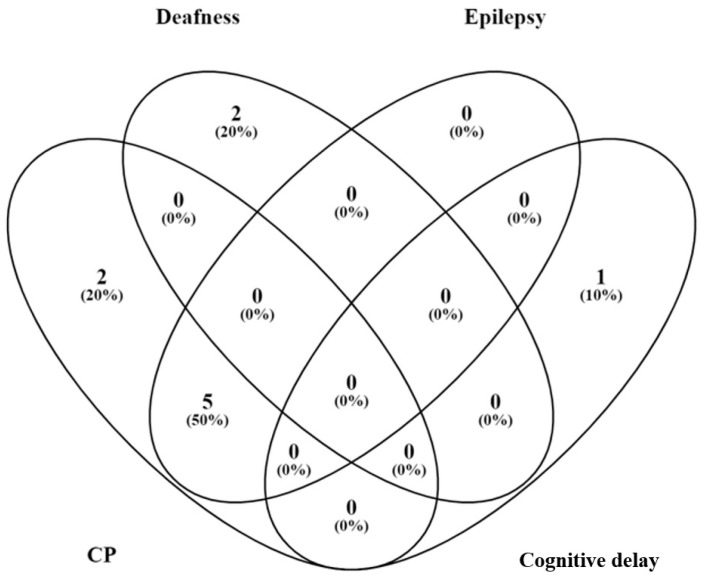
Venn diagram of infants with severe neurodevelopmental outcome. Patients with severe outcome are reported in the Venn diagram. CP: cerebral palsy.

**Figure 4 children-09-01194-f004:**
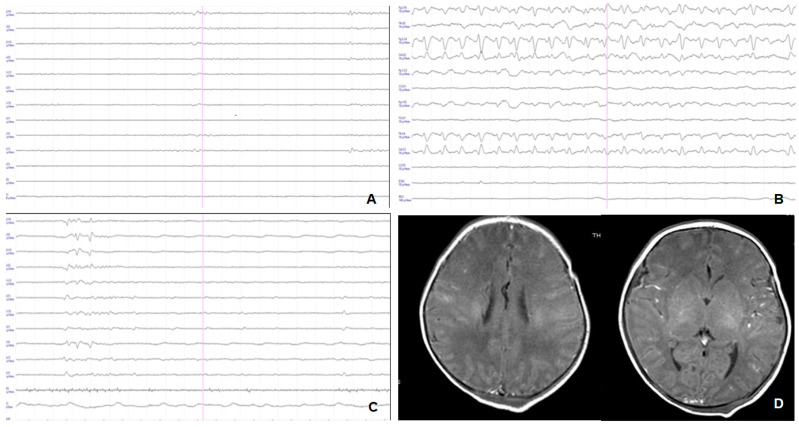
Patient 5 with severe HIE and severe neurodevelopmental outcome at 24 months of life. (**A**): p-EEG at enrollment (4 h of life) showing inactive EEG abnormalities (grade 4). (**B**): p-EEG at 12 h of life showing electrical seizures. (**C**): p-EEG at the end of TH showing severe EEG abnormalities (grade 3). (**D**): Cerebral MRI on day 5, showing pattern 1.

**Figure 5 children-09-01194-f005:**
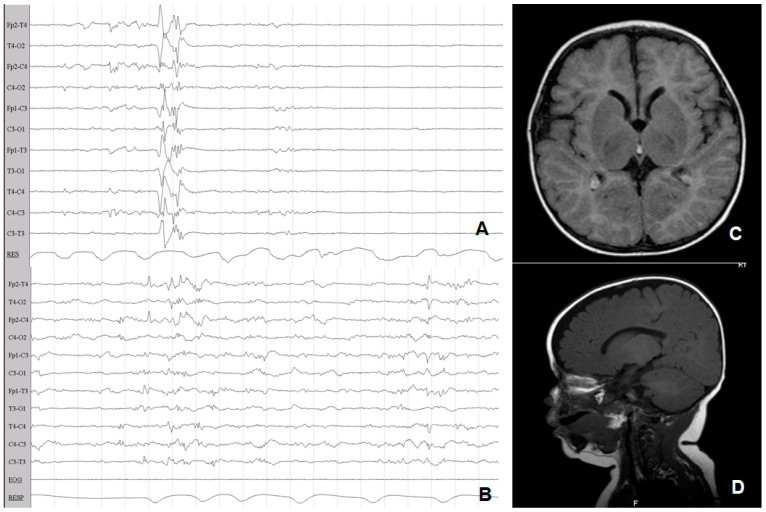
Patient 10 with severe HIE at enrollment and normal outcome at 24 months. (**A**): p-EEG at enrollment (3 h of life) showing severe EEG abnormalities (grade 3). (**B**): p-EEG at the end of TH showing mild EEG abnormalities (grade 1). (**C**,**D**): Normal cerebral MRI at 1 month of life.

**Figure 6 children-09-01194-f006:**
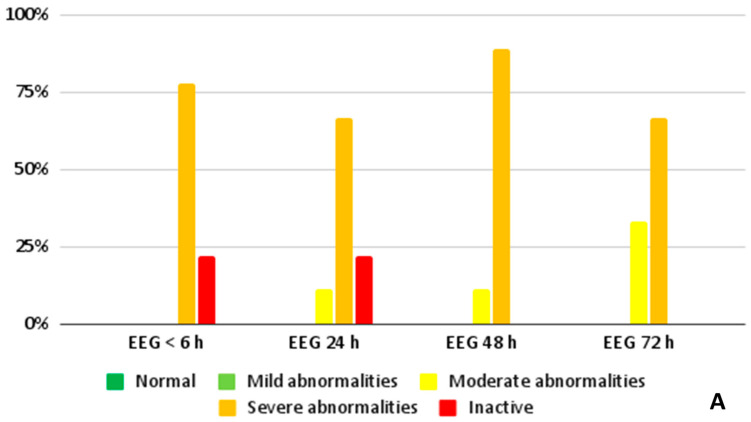
p-EEG variation in patients with severe (**A**) and normal or moderately abnormal neurdevelopmental outcomes (**B**).

**Table 1 children-09-01194-t001:** Comparison of cooled and un-cooled asphyxiated infants.

	All HIE (n = 139)	Cooled HIE (n = 82)	Un-Cooled HIE (n = 57)	*p*
Inborn Outborn	59 (42.4%) 80 (57.66%)	41 (50%) 41 (50%)	18 (31.6%) 39 (68.4%)	0.0470 *
Sentinel event Present Absent	114 (82.0%) 25 (18.0%)	62 (75.6%) 20 (24.4%)	52 (91.2%) 5 (8.8%)	0.0329 *
Delivery Vaginal Cesarean	91 (65.5%) 48 (34.5%)	52 (63.4%) 30 (36.6%)	39 (68.4%) 18 (31.6%)	0.6678
Weight	3381.30 ± 488.7	3379.9 ± 520.6	3383.3± 443.3	0.2030
Gestational age	39.66 ± 1.5	39.48 ± 1.41	39.9 ± 1.6	0.3240
Apgar 1st minute	2.48 ± 1.9	1.93 ± 1.55	3.3 ± 2.1	0.0090 *
Apgar 5th minute	5.06 ± 2.0	4.16 ± 1.74	6.3 ± 1.4	0.0001 *
Apgar 10th minute	6.52 ± 1.9	5.70 ± 1.74	7.7 ± 1.6	0.0001 *
pH	6.98 ± 0.2	6.92 ± 0.15	7.1 ± 0.1	0.5530
BE	15.96 ± 6.0	17.60 ± 6.08	13.6 ± 4.9	0.0001 *
HIE -Mild -Moderate -Severe	71 (51.17%) 45 (32.4%) 23 (16.5%)	14 (17.1%) 45 (54.9%) 23 (28.0%)	57 (100%) 0 (0%) 0 (0%)	<0.0001 *
p-EEG -Normal -Mild abnormalities -Moderate abnormalities -Severe abnormalities -Inactive p-EEG	10 (7.2%) 47 (33.8%) 43 (30.9%) 35 (25.2%) 4 (2.9%)	0 0 43 (52.4%) 35 (42.7%) 4 (4.9%)	10 (17.5%) 47 (82.5%) 0 0 0	<0.0001 *
Seizures -Absent -Present	109 (78.4%) 30 (21.6%)	52 (63.4%) 30 (36.6%)	57 (100%) 0	<0.0001 *
Cerebral MRI -Pattern 1 -Pattern 2 -Pattern 3 -Pattern 4 -Pattern 5	13 (9.4%) 11 (7.9%) 7 (5.0%) 23 (16.5%) 85 (61.2%)	13 (15.9%) 11 (13.4%) 2 (2.4%) 14 (17.1%) 42 (51.2%)	0 0 5 (8.8%) 9 (15.8%) 43 (75.4%)	0.0001 *
FM -Normal -Abnormal -Absent	108 (77.7%) 7 (5.0%) 24 (17.3%)	59 (72.0%) 3 (36.6%) 20 (24.4%)	49 (85.96%) 4 (7.2%) 4 (7.2%)	0.0237 *
Outcome -Normal -Moderately abnormal -Severe	122 (87.8%) 7 (5.0%) 10 (7.2%)	66 (80.5%) 7 (8.5%) 9 (11.0%)	56 (98.2%) 0 1 (1.8%)	0.0066 *
GMDS-R -DQ -Locomotor subscale -Eye & Hand Coordination subscale -Personal & Social subscale -Hearing & Language subscale -Cognitive Performance subscale	101.9 ± 15.6 100.8 ± 16.6 104.9 ± 16.8 97.4 ± 20.0 107.8 ± 15.4 101.8 ± 14.6	99.76 ± 17.9 97.72 ± 18.2 102.32 ± 18.2 95.30 ± 22.0 106.04 ± 17.9 100.64 ± 17.1	105.19 ± 10.8 105.26 ± 12.8 108.80 ± 13.7 100.26 ± 16.6 110.31 ± 10.2 103.42 ± 10.0	<0.001 * 0.0050 * 0.0250 * 0.0280 * 0.0010 * <0.001 *

Sentinel event: placenta abruption, umbilical cord prolapse, umbilical cord knot. BE: base excess. HIE: hypoxic ischemic encephalopathy. p-EEG: polygraphic electroencephalographic monitoring. MRI: cerebral magnetic resonance imaging. FM: fidgety movements, Griffiths Mental Developmental Scales: GMDS-R. Global Development Quotient: DQ. χ2 analysis was used for categorical variables. Analysis of variance and Mann–Whitney U tests were used for continuous variables.*: statistically significant.

**Table 2 children-09-01194-t002:** Neuro-developmental outcome of infants with HIE who underwent TH.

	Severe Outcome (n = 9)	Normal or Moderately Abnormal Outcome (n = 73)	*p*
Inborn Outborn	2 (22.2%) 7 (77.8%)	39 (53.4%) 34 (46.6%)	0.154
Sentinel event * Present Absent	2 (22.2%) 7 (77.8%)	18 (24.7%) 55 (75.3%)	0.028
Delivery Vaginal Cesarean section	5 (55.6%) 4 (44.4%)	47 (64.4%) 26 (35.6%)	0.017
Gestational age	39.8 ± 1.9	39.4 ± 1.3	0.1843
Weight	2993.9 ± 629.3	3427.5 ± 489.8	0.0908
Apgar 1st minute	1.6 ± 1.7	1.9 ± 1.5	0.4804
Apgar 5th minute	3.5 ± 2.1	4.2 ± 1.7	0.2826
Apgar 10th minute	5.0 ± 2.1	5.8 ± 1.7	0.3585
pH	6.8 ± 0.1	6.9 ± 0.1	0.1964
BE	21.1± 3.2	17.2 ± 6.2	0.0323 *
Encephalopathy severity -Mild -Moderate -Severe	0 1 (11.1%) 8 (88.9%)	14 (19.2%) 44 (60.3%) 15 (20.6%)	0.0002 *
p-EEG -Moderate p-EEG abnormalities -Severe p-EEG abnormalities -Inactive p-EEG	0 7 (77.8%) 2 (22.2%)	43 (58.9%) 28 (38.4%) 2 (27.4%)	0.0002 *
Seizures Absent Present	0 9 (100%)	52 (71.2%) 21 (28.8%)	0.0001 *
Cerebral MRI -Pattern 1 -Pattern 2 -Pattern 3 -Pattern 4 -Pattern 5	4 (44.4%) 3 (33.3%) 1 (11.1%) 1 (11.1%) 0	9 (12.3%) 8 (11.0%) 1 (1.4%) 13 (17.8%) 42 (57.5%)	0.0002 *
FM -Normal -Abnormal -Absent	1 (11.1%) 1 (11.1%) 7 (77.8%)	58 (79.5%) 2 (2.7%) 13 (17.8%)	<0.0001 *
GMDS-R -DQ -Locomotor subscale -Eye & Hand Coordination subscale -Personal & Social subscale -Hearing & Language subscale -Cognitive Performance subscale	65.7 ± 23.8 65.7 ± 24.5 63.9 ± 21.6 53.6 ± 10.3 74.7 ± 30.8 72.00 ± 27.6	103.9 ± 11.5 101.7 ± 12.8 107.1 ± 10.7 99.4 ± 18.3 109.9 ± 10.9 104.2 ± 11.3	<0.0001 * 0.0002 * <0.0001 * <0.0001 * 0.0029 * 0.0021 *

Sentinel event: placenta abruption, umbilical cord prolapse, umbilical cord knot. BE: base excess. HIE: hypoxic ischemic encephalopathy. p-EEG: polygraphic electroencephalographic monitoring. MRI: cerebral magnetic resonance imaging. FM: fidgety movements, Griffiths Mental Developmental Scales: GMDS-R. Global Development Quotient: DQ. χ2 analysis was used for categorical variables. Analysis of variance and Mann–Whitney U tests were used for continuous variables.*: statistically significant.

**Table 3 children-09-01194-t003:** p-EEG in patients with normal, moderate, and severe neurodevelopmental outcome.

	Normal Outcome (n = 66)	Moderate Outcome (n = 7)	Severe Outcome (n = 9)	*p*
p-EEG under age 6 h Normal p-EEG Mild p-EEG abnormalities Moderate p-EEG abnormalities Severe p-EEG abnormalities Inactive p-EEG	0 0 42 (63.6%) 24 (36.4%) 0	0 0 1 (14.3%) 5 (71.4%) 1 (14.3%)	0 0 0 7 (77.8%) 2 (22.2%)	<0.0001 *
p-EEG at age 24 h Normal p-EEG Mild p-EEG abnormalities Moderate p-EEG abnormalities Severe p-EEG abnormalities Inactive p-EEG	0 15 (22.7%) 42 (63.6%) 9 (13.6%) 0	0 1(14.3%) 2(28.6%) 4 (57.1%) 0	0 0 1 (11.1%) 6 (66.7%) 2(22.2%)	<0.0001 *
p-EEG at age 48 h Normal p-EEG Mild p-EEG abnormalities Moderate p-EEG abnormalities Severe p-EEG abnormalities Inactive p-EEG	0 34 (51.5%) 29 (43.9%) 3 (45.5%) 0	0 2 (28.6%) 1 (14.3%) 4 (57.1%) 0	0 0 1 (11.1%) 8 (88.9%) 0	<0.0001 *
p-EEG at age 72 h Normal p-EEG Mild p-EEG abnormalities Moderate p-EEG abnormalities Severe p-EEG abnormalities Inactive p-EEG	2 (3.03%) 52 (78.79%) 12 (18.2%) 0 0	1(14.3%) 2 (28.6%) 4(57.1%) 0 0	0 0 3 (33.3%) 6 (66.7%) 0	<0.0001 *

p-EEG: polygraphic electroencephalographic monitoring. χ2 analysis was used for statistical analysis. *: statistically significant.

**Table 4 children-09-01194-t004:** Uni- and multivariate analysis in cooled infants.

	Uni-Variate Analysis			Multivariate Analysis		
	OR	CI	*p*	OR	CI	*p*
p-EEG < 6 h	11.1	2.3–53.4	0.0025 *	-	-	-
p-EEG at 24 h	26.2	3.3–207.1	0.0019 *	-	-	-
p-EEG at 48 h	36.9	4.3–316.9	0.0010 *	27.6	2.8–267.5	0.0042 *
FM	5.0	1.9–13.0	0.0010 *	3.2	1.0–10.0	0.0475 *
HIE	24.2	2.9–202.5	0.0033 *	-	-	-
Cerebral MRI	0.4	0.2–0.7	0.0020 *	-	-	-
Apgar 1st minute	0.9	0.5–1.4	0.5476	-	-	-
Apgar 5th minute	0.8	0.5–1.2	0.2594	-	-	-
Apgar 10th minute	0.8	0.5–1.2	0.2286	-	-	-
BE	1.1	0.9–1.3	0.0892	-	-	-
PH	0.1	0.0–4.4	0.1489	-	-	-
Seizures	46.3	5.6–384.9	0.0004 *	-	-	-
Mode of delivery	1.5	0.4–5.9	0.6054	-	-	-
Inborn	4.0	0.8–20.6	0.0962	-	-	-
Sentinel event	0.9	0.2–4.6	0.8725	-	-	-
Sex	0.7	0.2–2.6	0.6067	-	-	-

BE: base excess. p-EEG: polygraphic electroencephalographic monitoring. MRI: cerebral magnetic resonance imaging. FM: fidgety movements, Griffiths Mental Developmental Scales: GMDS-R. Global Development Quotient: DQ. *: statistically significant.

**Table 5 children-09-01194-t005:** Prognostic accuracy for severe outcome in infants undergoing TH.

	Sensitivity % (95% CI)	Specificity % (95% CI)	PPV %	NPV %	ROC (95% CI)
HIE (criterion: >moderate)	88.9 (51.7–98.2)	79.5 (68.4–88.0)	34.8	98.3	0.85 (0.7–0.9)
p-EEG < 6 h (criterion > 2)	100 (66.2–100)	58.9 (46.8–70.3)	23.1	100	0.83 (0.7–0.90)
p-EEG 48 h (criterion > 2)	88.9 (51.7–98.2)	90.4 (81.2–96.0)	53.1	98.5	0.92 (0.8–1)
Seizure (criterion: present)	100 (66.2–100)	71.2 (59.4–81.2)	30	100	0.85 (0.8–0.9)
Cerebral MRI pattern (criterion: pattern ≤ 3)	88.9 (51.7–98.2)	75.3 (63.9–84.7)	30.8	98.2	0.84 (0.7–0.9)
FMs (criterion: abnormal or absent)	88.9 (51.7–98.2)	79.5 (68.4–88.0)	34.8	98.3	0.84 (0.7–0.9)

HIE: hypoxic–ischemic encephalopathy. p-EEG: polygraphic electroencephalographic monitoring. MRI: cerebral magnetic resonance imaging. FMs: fidgety movements. PPV: positive predictive value. NPV: negative predictive value. ROC: receiver-operating characteristic).

**Table 6 children-09-01194-t006:** Comparison of cooled and un-cooled neonates with mild HIE.

	All Mild HIE (n = 71)	Un-Cooled Mild HIE (n = 57)	Cooled Mild HIE (n = 14)	*p*
Weight	3373.9 ± 426.7	3383.3 ± 443.3	3335.9 ± 363.3	0.8003
Gestational age	39.8 ± 1.6	39.9 ± 1.6	39.5 ± 1.4	0.3550
Inborn Outborn	46 (64.8%) 25 (35.2%)	39 (68.4%) 18 (31.6%)	7 (50.0%) 7 (50.0%)	0.3267
Sentinel event Present Absent	9 (12.7%) 62 (87.3%)	5 (87.7%) 52 (9.2%)	4 (18.6%) 10 (71.4%)	0.1219
Delivery Vaginal Cesarean	50 (70.4%) 21 (19.6%)	39 (68.4%) 18 (31.6%)	11 (78.6%) 3 (21.4%)	0.6753
Apgar 1st minute	3.1 ± 2.1	3.3 ± 2.1	2.3 ± 1.6	0.1837
Apgar 5th minute	5.9 ± 1.9	6.3 ± 1.6	4.5 ± 2.1	0.0055 *
Apgar 10th minute	7.4 ± 1.6	7.7 ± 1.4	6.2 ± 1.9	0.0103 *
pH	7.1 ± 0.2	7.1 ± 0.2	7.0 ± 0.1	0.5019
BE	13.7 ± 4.9	13.6 ± 4.6	13.6 ± 4.9	0.9821
p-EEG -Normal -Mild abnormalities -Moderate abnormalities -Severe abnormalities -Inactive p-EEG	10 (14.1%) 47 (66.2%) 10 (14.1%) 4 (4.6%) 0	10 (17.5%) 47 (82.5%) 0 0 0	0 0 10 (71.4%) 4 (18.6%) 0	0.001 *
Seizure Absent Present	71 (100%) 0	57 (100%) 0	14 (100%) 0	-
Cerebral MRI -Pattern 1 -Pattern 2 -Pattern 3 -Pattern 4 -Pattern 5	0 0 5 (7.0%) 11 (15.5%) 55 (77.5%)	0 0 5 (8.8%) 9 (15.8%) 43 (75.4%)	0 0 0 2 (14.3%) 12 (85.7%)	0.4972
Fidgety Movements Normal Abnormal Absent	61 (85.9%) 5 (7.0%) 5 (7.0%)	49 (86.0%) 4 (7.0%) 4 (7.0%)	12 (85.7%) 1 (7.1%) 1 (7.1%)	0.9997
Outcome -Normal -Moderately abnormal -Severe	70 (98.6%) 0 1 (1.4%)	56 (98.2%) 0 1 (1.8%)	14 (100%) 0 0	0.4434
GMDS-R -DQ -Locomotor subscale -Eye & Hand Coordination subscale -Personal & Social subscale -Hearing & Language subscale -Cognitive Performance subscale	104.7 ± 11.2 104.9 ± 12.0 108.5 ± 13.4 99.5 ± 17.8 109.5 ± 11.5 103.6 ± 11.4	105.2 ± 10.8 105.3 ± 12.8 108.8 ± 13.7 100.3 ± 16.6 110.3 ± 10.2 103.4 ± 16.4	102.6 ± 13.1 103.6 ± 8.4 107.3 ± 12.6 96.4 ± 22.6 106.0 ± 15.7 104.3 ± 19.9	0.3819 0.7781 0.3899 0.7746 0.2418 0.8737

BE: base excess. HIE: hypoxic ischemic encephalopathy. P-EEG: polygraphic electroencephalographic monitoring. MRI: cerebral magnetic resonance imaging. FMs: fidgety movements, Griffiths Mental Developmental Scales: GMDS-R. Global Development Quotient: DQ. χ2 analysis was used for categorical variables. Analysis of variance and Mann–Whitney U tests were used for continuous variables. *: statistically significant.

## Data Availability

Data are available upon request to the corresponding Authors.

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
