# Peer review of "Polygraphic EEG Can Identify Asphyxiated Infants for Therapeutic Hypothermia and Predict Neurodevelopmental Outcomes"

_children, 2022, doi:10.3390/children9081194_

Round 1

Reviewer 1 Report

we read with interest the article by Lugli et al titled Polygraphic EEG can Identify Asphyxiated Infants for Therapeutic Hypothermia and Predict Neurodevelopmental Outcomes

The authors evaluated the neurodevelopmental outcomes in surviving asphyxiated neonates who did not receive therapeutic hypothermia (TH) based on clinical and polygraphic electroencephalographic (p-EEG) criteria.

This is a well-designed studies where the authors have collected data on prolonged p-EEG within during TH. of interest, p-EEG has shown high diagnostic and prognostic accuracy by correctly selecting neonates to undergo TH and identifying those with poor outcomes. the work would be of excellent blood biomarkers that can be complementary to the p-EEG data and that can highlight if p-EEG can corelate dto well established blood biomarkers as many studies have established that these infants would suffer Neonatal brain injury.

it would be optimal to look for inflammatory markers (Il-1, IL-10, TNF-alpha, HIE and or brain injury biomarkers such as UCH-L1 and GFAP)

 This can enrich the data value presented in the well-designed work.

If possible to conduct, i highly encourage this to be performed or at least there should be a section on current serum biomarkers in these patients and can be added as limitations

Author Response

Dear Reviewer 

thenk you for your comments. We have modified the manuscript according to your suggestions. You find enclosed the file with response to your comments.

Best regards

Licia Lugli

Reviewer 2 Report

Lugi et al.  manuscript entitled "Polygraphic EEG can Identify Asphyxiated Infants for Therapeutic Hypothermia and Predict Neurodevelopmental Outcomes", where the authors evaluated asphyxiated neonates with/without therapeutic hypothermia (TH) intervention towards the prediction of the neurodevelopmental outcomes. The authors utilizes polygraphic EEG signature in the human neonates and determined that severe neurodevelopmental disability occurred in less than 10 % confirming the protective role of TH. Interestingly, the authors were able to get the best predictions when p-EEG evaluations conducted at 48h along with general movement.

The strength of the article is that it was able to characterize the HIE grade and moderate to severe p-EEG abnormalities.

However, the weakness of  the article is as follows.

Abstracts : A better logical flow and smooth transition is needed.
- Please start with the definition of Neonatal encephalopathy? What is the primary factor that causes this type of phenomenon or symptoms? What is unknown in the field? The severity of the diseases etc.
-Line 23: "57 who did not"- please rearrange the sentences and try to explain it with better writing styles.
-Line 32: The conclusion part of the abstract should be coherent with the earlier text. Please highlight what are the novel findings of this study which helps to move the field forward.

Introduction section:
Line 39: Please also discuss which brain regions are more prone to get affected first during perinatal asphyxia. Also, please discuss the brain protective mechanisms such as BBB's role during  asphyxia related insult.

Method section:
Line 117-119: Please make sure that the text in the Figure 1 is readable or printable. You might need to increase the font size.
If you have used the scientific machines, please provide the details of those machine used (company name, model, capacity etc).

Results section:

I encourage the authors to provide representative experimental data from their own experiment wherever possible. Actually all those Tables can be a part of supplemental figures. The readers find more originality when one provides the brain scans or other traces obtained directly from their equipments for comparative study.

 Line 182-185: The text below those Table should be elaborative. For example, what statistical test were used in the calculation above in the Table and  what p-values are considered significant here, please mention. You can also place asterisk on  the data that showed significant different.

If you find  way to plot those quantified data, that will make it simple to understand for readers.  For example, pie chart, heat map or any other appropriate plot will be helpful (Consult the statistician, if you possible)

Discussion :

Line  329-330: Scheme 1: Is the scale bar/magnification same for all those traces? Please provide the legend. You can replace "Scheme" with "Figure". You do not need to mention 1A or 1B, Just mention A, B, C, D in the legends without numerical.

I was expecting you to provide similar traces for the figures in Result section of your manuscript rather than filling this space with someone else' work occupying big space. Have you taken permission to adopt this figure here from the authors?

Line 334-335: There are multiple venn diagrams here. I would make it colorful and explain what we are observing from this Scheme 2 in details in the legend. Please be clear whether these Figure arised from your own experiment or other's work. If it is from your own work, I encourage you to place it in the Result section. If you are predicting or trying to sumarize your studies with unknown gaps, then it it can still find the  place here otherwise somewhere else !

Note that both Scheme 1 and Scheme 2 would fit much better in the discussion sections rather than placing them prior to conclusion part.

Conclusions: Line 334: The conclusion section should be started in a separate paragraph.

 References: Line:347-437: Please check the references consistently follow the journal's guidelines. Omit "month", underline" or other inconsistencies.

Overall, the article at the present state needs to work more on presentation style by replacing most of the tabular data with the original figure plots. Logical flow and addressing the issues by adding missing statistical tests used in the appropriate text (p-values, n, statistical test used) should come side by side in most of the time in the text. The author should also highlight what are the novel findings they specifically observe in this article that can move the field forward.

Author Response

Dear Reviewer

Thank you for your comments. We have modified the manuscript according to your suggestions. Please find enclosed the file with response to your comments.

Best Regards

Licia Lugli
